# Drivers or Pedestrians, Whose Dynamic Perceptions Are More Effective to Explain Street Vitality? A Case Study in Guangzhou

Yuankai Wang [1,2], Waishan Qiu [3], Qingrui Jiang [1,2], Wenjing Li [4], Tong Ji [5] and Lin Dong [6,*]

1   Bartlett School of Architecture, University College London, 22 Gordon St., London WC1H 0QB, UK
2   College of Architecture and Urban Planning, Tongji University, 1239 Siping Road, Shanghai 200092, China
3   Department of City and Regional Planning, Cornell University, Ithaca, NY 14853, USA
4   Center for Spatial Information Science, The University of Tokyo, Tokyo 113-8654, Japan
5   China Energy Engineering Group, Zhejiang Electric Power Design Institute Co., Ltd.,
    Hangzhou 310003, China
6   School of Landscape Architecture, Nanjing Forestry University, Nanjing 210037, China
*   Correspondence: dongl@njfu.edu.cn

**Abstract:** As an important indicator of urban development capacity, vitality can be affected by the human perception of street views, which is a dynamic sensory process that can differ greatly according to different transportation modes, due to their different travel speeds, distances, and routes. However, few studies have evaluated how the dynamic spatial perceptions differ between different travel modes and how these differences can affect vitality differently, due to the limitation of city-scale quantitative data on the dynamic perception of urban scenes. To fill the gap, we propose a "dynamic through-movement perception" (DTMP) measure which integrates a streetscape quality evaluation model with a network-based movement potential model. We measure the streetscape qualities from Baidu street-view images (SVI) and compare the spatial perceptions of drivers and pedestrians in central Guangzhou, China. First, more than twenty visual elements were classified from SVIs to predict human perceptions collected from visual surveys. Second, the through-movement probability of driving and walking were calculated based on classic natural movement theory in space syntax and measured as the angular betweenness for the two travel modes. Third, we accumulate the multipliers of visual perception and through-movement probability of driving and walking as the DTMP for both modes. Lastly, the DTMPs of both modes were fitted into linear regression models to explain street vitality, which is measured using Baidu mobile phone check-in data, when other control variables such as functional density, functional diversity and amenity clustering reachability are accounted for. The results show that the dynamic perception of driving overall shows a stronger correlation with street vitality, while perceived richness is significantly positive in both travel modes. This study provides the first quantitative evidence to reveal how the movement probability of different travel modes can significantly influence people's sense of place, while in turn increasing street vitality. Our results can explain how different types of street commerce (i.e., pedestrian-oriented, and auto-oriented) aggregate spontaneously due to the dynamic movement potential, which provides an important reference for urban planners and decision makers for improving street vitality when making urban revitalization policies.

**Keywords:** street vitality; dynamic perception; travel modes; network betweenness; street view image

## 1. Introduction

### 1.1. Context

Human perception is very important for the aggregation of urban street vitality. However, how human perception can differ according to different travel modes has been discussed very little. In this paper, (1) we hypothesize that the perception of a place is a dynamic process influenced by different modes of transportation; (2) we innovatively

propose a network-based model to test which of the dynamic perceptions, driving or walking, can better depict the street vitality of Guangzhou.

Vitality includes both spatial vitality and socio-economic vitality. As one of the most used spatial indicators, street vitality often refers to the frequency of activities taking place in the street. It is one of the most important factors for evaluating the comprehensive quality of a region, while greater vitality can promote the value of surrounding residential units and industrial clusters [1,2]. Prior studies revealed that street vitality is related to its physical environment in many ways.

On the one hand, the formation of vitality is a bottom-up process regarding how people decide where to go, work and live, decisions that follow their perceived sense of place [3–6]. For example, people prefer to gather and stay in places they perceive better [7,8], which in return induces higher travel flow volumes [9]. Consequently, more shops and business would be attracted these areas spontaneously [10]. That said, better street vitality would attract more people to conduct social activities, thus fostering even more population flows [9], and improving the perceived safety [11]. At the same time, the attractiveness of a place also generates more human dynamics, which in return induces more commercial demands, which results in improved street vitality in both quantity and quality, as business and shops often follow the population flows [12–17]. On the other hand, it is also a top-down process. Local governments and real-estate developers will also see opportunities and implement urban renewal projects in regions that have a better sense of place [18,19], which makes the streets more attractive, inducing more population flows [1,20].

Therefore, street vitality is interwoven with sense of place. They can both promote the growth of regional value (Figure 1). Understanding the connection between sense of place and street vitality can better predict the human dynamics and the evolving of a city, to assist local governments, property investors and street vendors to identify potential areas for growth.

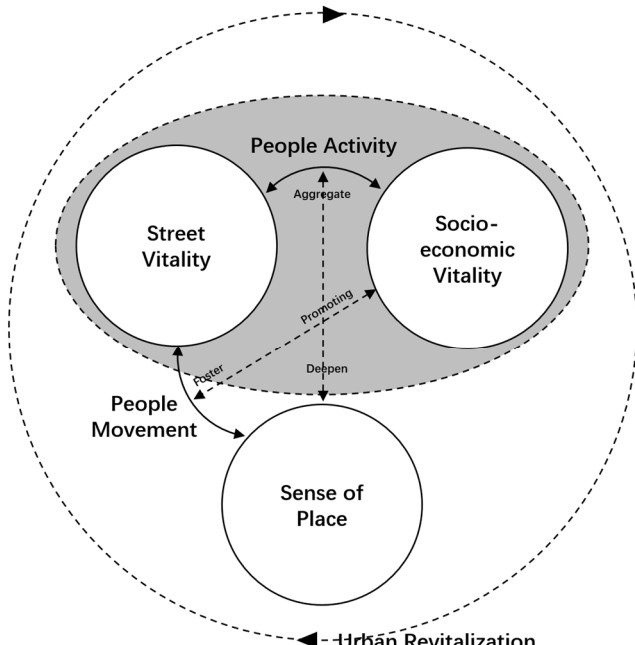

**Figure 1.** The circle of Street vitality.

## 1.2. Research Gap

The recent prevalence of geolocation activities data [21] and street view images (SVI) [22] has enabled the measurement of urban scale street scene appearance at a fine-grained resolution [18], encouraging researchers probing into urban vitality from a more human-scale perspective. Emerging studies tried to link urban vitality to spatial perception from a human visual perspective through the segmentation of SVI [23,24]. For example, Qiu et al. [6] evalu-

ated the large-scale urban perceptions subjectively and objectively. Kang et al. [5] used SVI to enrich the explanatory power of housing prices. Salazar Miranda et al. [25] captured how built environment influences the walking routes.

However, how human perception can differ according to different travel modes, which may ultimately affect urban vitality in different ways, has been discussed very little. Specifically, human perception is a dynamic process [26] rather than a static or fixed result [11,27]. First, sense of place is a continuous, dynamic, and comprehensive process largely affected by the ways how a street user accesses and experiences the destination place. During the process, when the travel speed, distance, and route vary, the street environment associated with and accumulated along the process are essentially different, which ultimately causes differences in the perceived quality of that place. That said, perception can differ according to an observer's individual differences as well as according to different modes. For example, people driving or taking public transportation may have a different sense of place compared to those who choose to walk or cycle. However, the difference of perception under the different movement modes, and how they relate to street vitality, have never been discussed. Second, which perception, driving or walking, is more effective in explaining street vitality in the global and local scale is still unknown. We hypothesize that for walkable urban areas, the walking perception would exhibit stronger powers of prediction for vitality, and vice versa. Third, few studies have examined the dynamic change of vitality in time and space at urban scale, while the empirical evidence linking the dynamic spatiotemporal change of vitality with dynamic human perception (due to the change of built environment) is missing.

### 1.3. Research Question & Hypothesis

To bridge this gap, we hypothesize that spatial perception is a dynamic process affected by different transportation modes, resulting in different powers of prediction for vitality. To test this, we propose a dynamic perception model which integrates the visual-based streetscape quality measurement with the network-based through-movement probability measurement (TMPM), which is a scientific modelling method that analyzes pedestrian, cycle and vehicle movement networks and has been widely studied in planning and design research. We expect that the perception received from different travel modes would have different positive and negative effects on people's preferences for a place [2], such that street vitality in some regions is largely affected by perception through walking while vitality in other areas is dominated by driving. The results of our research design will provide empirical evidence to examine the efficiency of the space use based on pedestrian-oriented or auto-oriented development strategy. For example, it will verify whether a pre-planned auto-oriented commercial business district has a better driving perception and a more positive contribution to urban vitality.

## 2. Literature Review

### 2.1. Lack of Dynamic Measure of Perception

Although prior studies on place-based perception have indicated that people's perception of place can be influenced by time and experience [28–30], or can fluctuate according to the route of one's connection to a place [31–35], most prior studies were still based on the static frameworks. That is, the measurement of perception is stationary and lacks inner relation (i.e., there is a missing overall relationship that connects each individual visual perception measured from each SVI frame). For example, Ewing and Handy (2009) [36] quantified five static perceptions, namely, imageability, enclosure, human scale, transparency, complexity from manually measuring built environment attributes such as the proportion of sky view, greenery and building using video clips of streetscapes. More recently, emerging studies in urban-scale street scene understanding started to use open source image data (e.g., Google Street View) and computer vision techniques (e.g., image semantic segmentation) to boost the pipeline [6,19,22].

However, the above methods are highly limited when it only evaluates the static perception where the street view image itself is located [37]. This paper argues that static and node-based scene evaluations are not capable of capturing the real-life sensory experience. They simply assume that each street scene has equal probably in being perceived, and largely neglect the deviation of the dynamic impact of spatial continuity and the subtle change on people's sense of place through the different linkage of street scenes. A previous study has proved that the appearance of the built environment (e.g., density, diversity and design) [18] affects the route and travel mode that people choose to take [38]. Therefore, each street scene was not able to be equally perceived.

That said, the dynamic participation of moving through a space should also be considered [26]. What scenes people see along the route when getting to the destination matters to how they perceive the quality of the destination [39]. For example, the dynamic green exposure when traveling can neutralize people's perception of actual greenness in a community with low green coverage [40].

As the relative spatial continuity and the contrast of the "route to a place" will affect people's perception of the place, the actual sense people perceive can vary according to different means of transport. For instance, people who drive a car and who choose slow-moving methods of transport such as walking/cycling may have a completely different sense of place in the same space. Moreover, it is well known that the aggregate flows [41], route choice [38] and other spatial phenomena are affected by the street network such as the way how a space being connected in the network system will deepen the perceived impression of that space [41–43]. For example, due to their different travel distances and speeds, pedestrians and drivers can have different route preferences for the same origin-destination decision [17,44–47], which can lead to different perceptions of the same destination, as the user would see different scenes along the routes.

## 2.2. Lack of Comparison between Dynamic Perceptions by Modes

The relationship between streetscape and sense of place can be moderated by dynamic traffic modes. Research in this regard has been limited to theories, lacking quantitative and empirical evidence due to the difficulties in measuring the dynamic perceptions according to different modes. Conventionally, it requires long-term observations using surveys and interviews, which is costly in time and money. Some studies started to take advantage of open-source data, crowd sourcing and artificial intelligence. For example, Ye et al. [48] measured the daily accessible street-view greenery by integrating people's accumulated senses on the daily accessible path. However, this study only investigated the non-parametric through-movement accessible green with two levels (as low and high) and ignored how different traffic modes would affect accumulated human perceptions. Only recently, Wang et al. [40] modeled dynamic and static greenness exposure for multi-traffic modes including walking, biking, e-biking and driving. However, their method used less-available GPS tracking data and costly questionnaires, which can hardly be generalized to the urban scale for other regions. This study noted that one alternative is using space syntax, which has been a popular measurement for street connectivity in the case of daily accessible green [48,49]. Space syntax can models not only the possible route choice of different travel modes, but also a comparison of the probability of space visited (i.e., through-movement) by different travel distances due to mode choice deviations. Both features play important roles in affecting people's realistic and overall perception of a destination. Hence, this study hypothesizes that by integrating human perception with through-movement models, a dynamic and network-based perspective that considers how multiple traffic modes can explain street vitality more comprehensively and effectively.

## 2.3. Vitality Can Be under the Influence of Dynamic Perception as Well

Many studies have proved that consumption places and point of interests (POI) such as cafes can explain most of the urban vitality distributions [50–54]. As a subset of urban vitality, "street vitality" often exhibits multi-dimensional characteristics due to the dynamic

nature of commercial activities [55] in time and space [3]. Therefore, street vitality can manifest its space-time dynamics according to the different through-movement probability of travel modes.

In light of the difficulty of measuring vitality and urban form, most research investigated the spatial features of the built environment such as the density of street geometries [12,18,56–58]. Although they found significant correlations between vitality and urban form, they were rather result-oriented or backward-thinking models [59], lacking theoretical foundation of human behaviors. The fundamentals of urban vitality is constantly related to social interaction [54] and human behaviors, though its definition has evolved. As Jacobs [60] claimed, the source of a vibrant street depends on its interaction with human perception [54].

Emerging studies have begun to explore the correlations between human perception and vitality [2]. For example, using the behavioral status of people in the scene as a measurement of commercial vitality, Li et al. [10] operationalized the correlations between vitality, perception and the street environment. However, in measuring perception, prior studies often ignored its multi-dimensional nature [61–63]. Few studies have investigated how dynamic perceptions perceived by people in the process of street exploration affect street vitality.

That said, the connection of a dynamic sense of place and street vitality under the influence of the different traffic modes should be addressed. It will provide an important reference to urban designers regarding how to curate a better sense of place for urban dwellers using different travel modes. It will also illuminate design interventions that can be instrumental in promoting better streetscape and street network topology, which ultimately improves local and regional vitality. In addition, the comparison between vitality and dynamic perception received by drivers versus pedestrians can generate important knowledge for re-positioning the auto-oriented city and create better urban landscapes for daily life [64]. In short, different traffic modes may play a significant role in determining the dynamic sense of place and street vitality. Therefore, this study set out to test how people's sense of place is affected when the static streetscapes extracted from SVI data are accumulated by the dynamic through-movement probability of different travel modes estimated from space syntax.

## 3. Data and Method

### 3.1. Study Area

With a total area of 7434 km², Guangzhou is situated at the heart of Guangdong Province and is the most populous built-up metropolitan area in Mainland China [65]. Its central area (788 km²) is selected as the case study (Figure 2) because it is one of the largest urban agglomerations serving the most vibrant street life in the Pearl River Delta metropolitan region [66]. This "mega city" has experienced an accelerated urbanization process in the past few decades [67], with the intensified urban space expansion, continuous growth of inhabitants and commercial and manufacturing regions brought by the booming economy [67]. However, this exponential urbanization growth has resulted in urban planning challenges such as traffic problems and issues of insufficient provision of amenities and consumption resources, which may inform the consequences of urban revitalization policy on urban design to encourage a more balanced living environment.

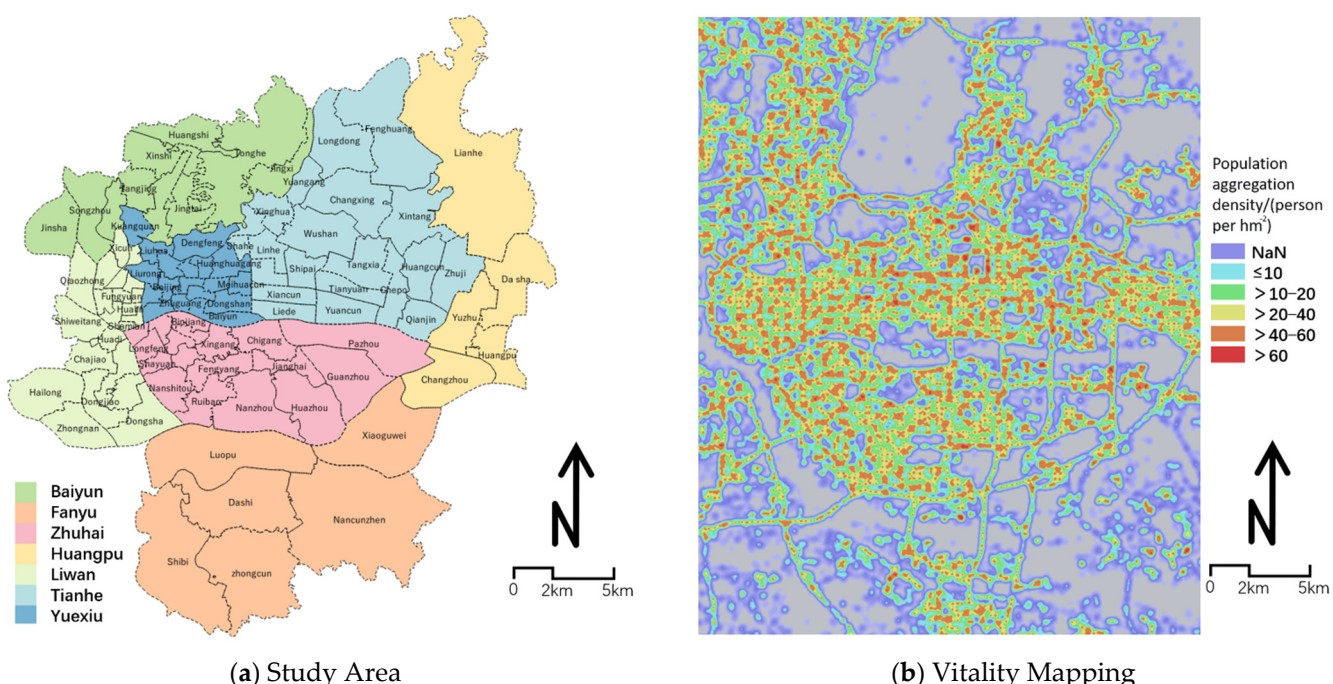

(**a**) Study Area         (**b**) Vitality Mapping

**Figure 2.** Study area (**a**) central urban area of Guangzhou and (**b**) its vitality in terms of mobile phone hotspot locations.

### 3.2. Analytical Framework and Data

The analytical framework is four-fold (Figure 3). First, we calculated subjective environmental perceptions (i.e., accessibility, aesthetics, enclosure, environmental greenness, environmental richness and human scale) within the study area from the Baidu Street View API (http://api.map.baidu.com/Ibsapi/, accessed on 1 April 2022) using subjective perception prediction methods [6,68]. Second, we calculated the TMPM for each street segment of the two travel modes (i.e., driving and walking) using space syntax. Third, we multiplied TMPM with the static perception data to form the new dynamic perception variable with movement probability weighting, which became our variable of interest. Third, we retained the classical urban functional indicators such as functional density, diversity, and accessibility in urban modelling as the control variables of each model. Fourth, we used the Baidu hotspot data as street vitality (i.e., the dependent variable), and fitted all independent variables. Notably, we conducted OLS regressions and compared their results to comprehensively compare how the dynamic street perception brought by walking versus driving is related to urban vitality globally.

There are four clusters of datasets: (1) Baidu SVI to measure streetscape perceptual index, (2) road network shapefile to measure through-movement probability route choice of walking and driving, (3) points of interest (POI) data in 2021 from Amap (https://lbs.amap.com/, accessed on 18 May 2022) to measure street-based functional accessibility and (4) the Baidu activity hotspot data to measure vitality measurement. Streetscape perception data was computed from the Baidu SVI with semantic segmentation algorithm. Through-movement probability route choice of walking and driving were calculated through the space syntax betweenness index with two modes of search radius, respectively. Street-based functional accessibility measured the spatial distribution of functional attractiveness by different types of POIs data. The activity hotspot data for measuring vitality was accumulated using the frequency of mobile phone check-in data by Baidu.

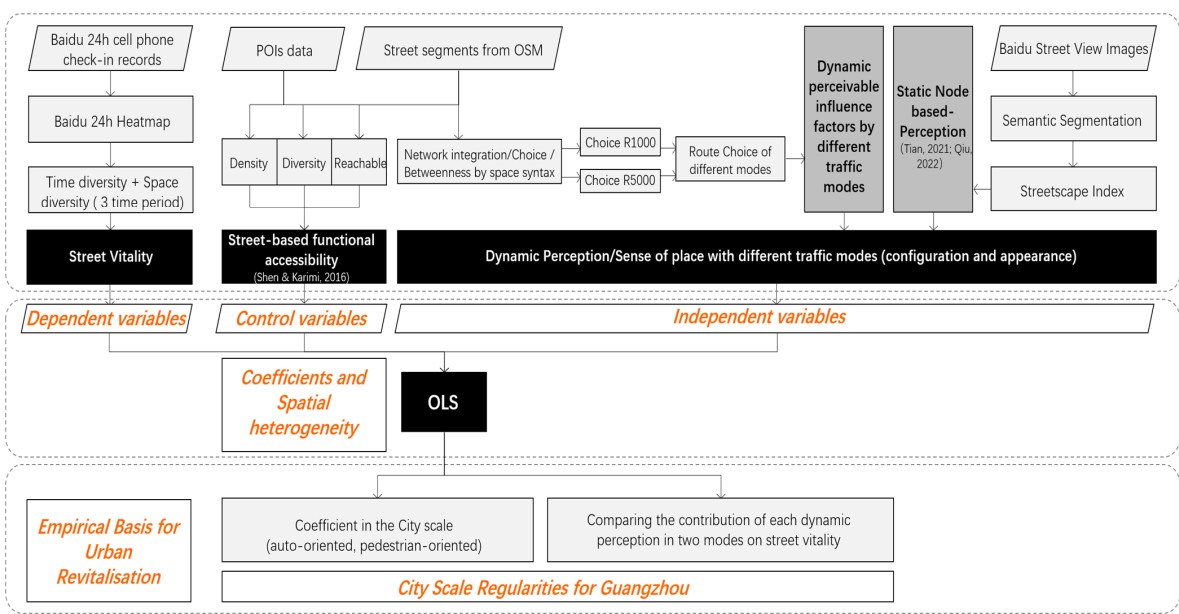

**Figure 3.** The analytical framework [6,68,69].

The first two variables are the variables of interest of this study, the third cluster of variables are the controlled variables, while the last is the dependent variable (Figure 3).

### 3.2.1. Streetscape Perceptual Measurements from Street-View Images

SVIs were downloaded from Baidu Street View Static API (http://api.map.baidu.com/lbsapi/, accessed on 1 April 2022) with fixed camera settings (Figure 4b). What Baidu SVI captures is a horizontal view of the physical environment in detail, which is close to the vision of pedestrians and drivers, and which could be useful to proxy human perception in the street [4,6].

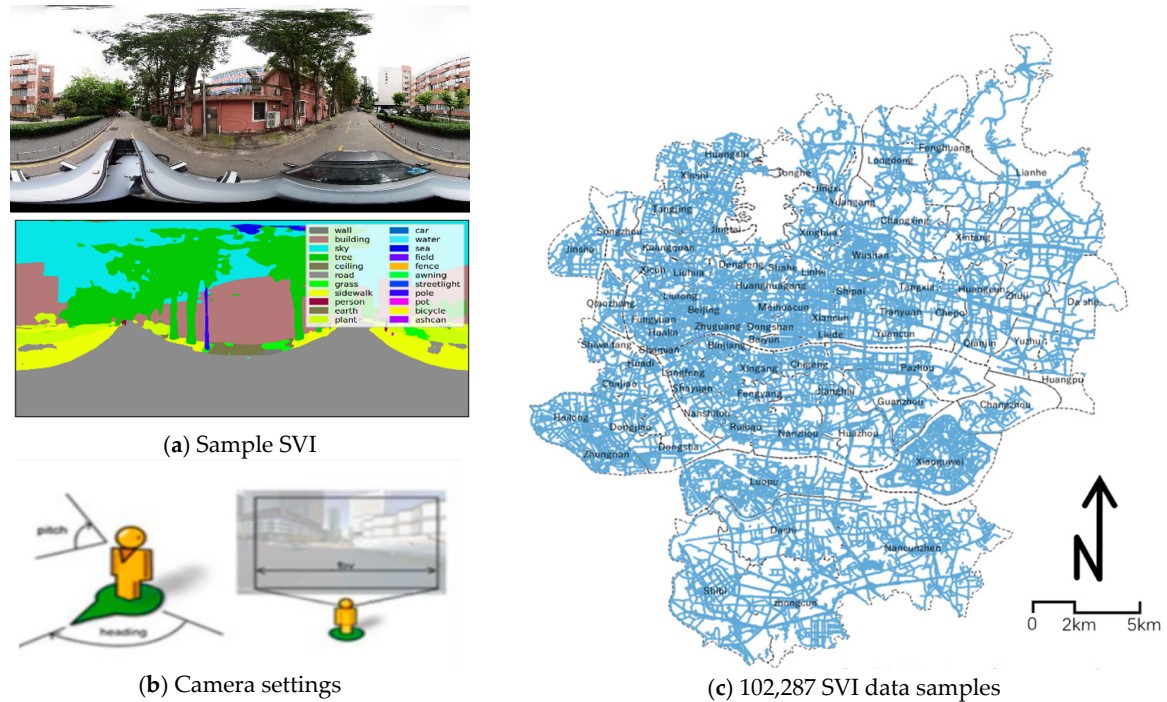

(**a**) Sample SVI

(**b**) Camera settings　　　　　　　　　　(**c**) 102,287 SVI data samples

**Figure 4.** Sampling SVI and semantic segmentation (**a**) Sample SVI. (**b**) Camera settings diagram. (**c**) 102,287 SVI training samples.

First, we sampled SVIs every 50 m [2,6,19,68] with 120 degrees for the horizontal field of view (FOV) and 0 degrees for the 'pitch' [6,68] of the camera along the street segments. In total, we obtained 102,287 images with 640 × 360 pixels of each size in central Guangzhou (Figure 4c).

Second, we classified the physical feature index by measuring the pixel ratio of a streetscape object to the total pixels of an image. To obtain the view index (e.g., building, sky, tree, sidewalk, grass, plant, road, signboard, etc.), we used a pre-trained semantic segmentation framework–Pyramid Scene Parsing Network (PSPNet) with the ADE20K dataset, which collected the data in 50 cities in different seasons and can predict 150 object classes of physical features in streetscape.

Third, we followed the method proposed by Tian [68] and Qiu [6], using 300 subjective SVIs survey to predict perception for all other unranked SVIs [6,68]. The accuracy of the model reached over 0.65 and the reliability of the prediction was proved in the case of Shanghai and Berlin. We then used 300 SVIs with labelled scores for predicting 7 perceptions of the urban environment that were referred to Ewing and Handy [36], namely: sense of order, accessibility, aesthetics, enclosure, environmental greenness, environmental richness and human scale.

Lastly, we achieved 102,287 perception scores with geolocation in total to describe the sense in central Guangzhou (Figures 5–7) (Table 1).

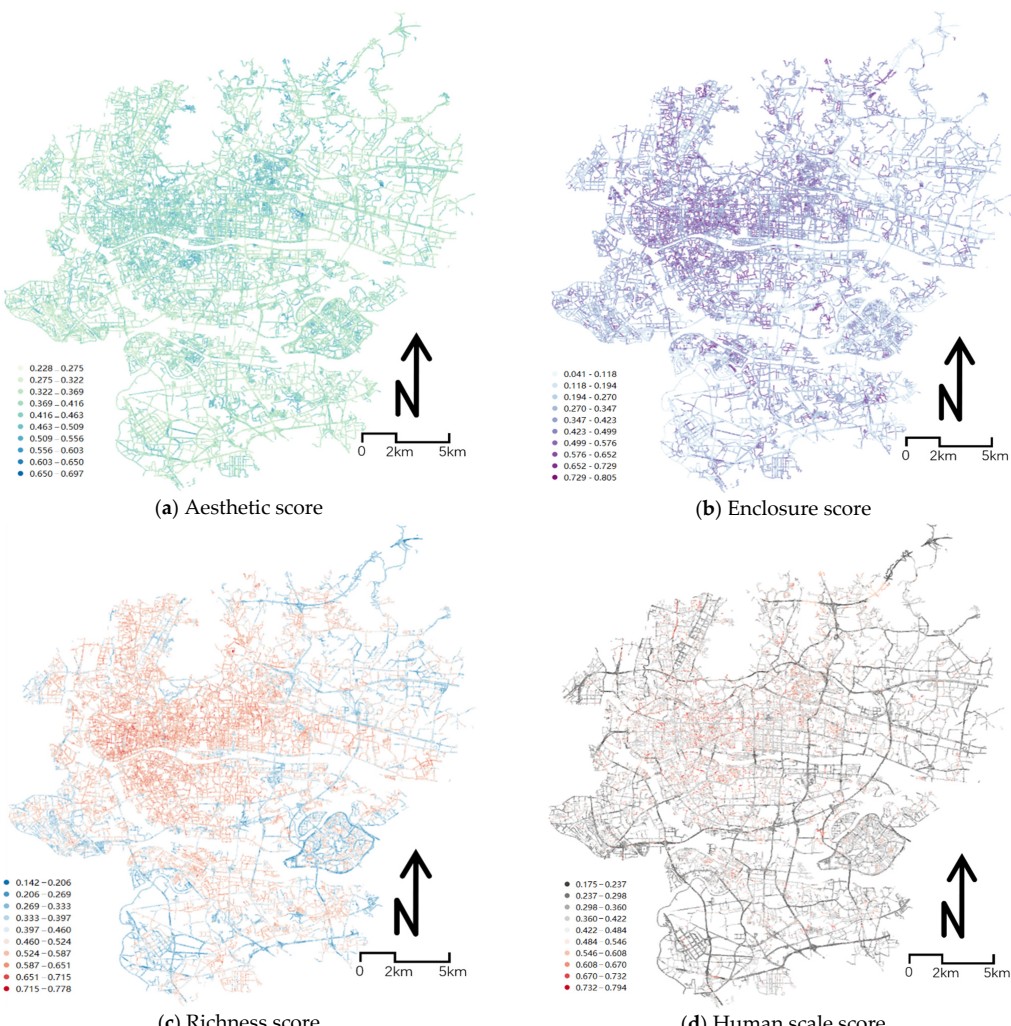

**Figure 5.** The spatial distribution of selected streetscape perception indices.

| Perception | SVI (Good) | SVI (Bad) | Histogram |
|---|---|---|---|
| (**a**) Order | | | |
| (**b**) Accessibility | | | |
| (**c**) Aesthetic | | | |
| (**d**) Eco | | | |
| (**e**) Enclosure | | | |
| (**f**) Richness | | | |
| (**g**) Human scale | | | |

**Figure 6.** Data distribution of 7 types of Streetscape perception.

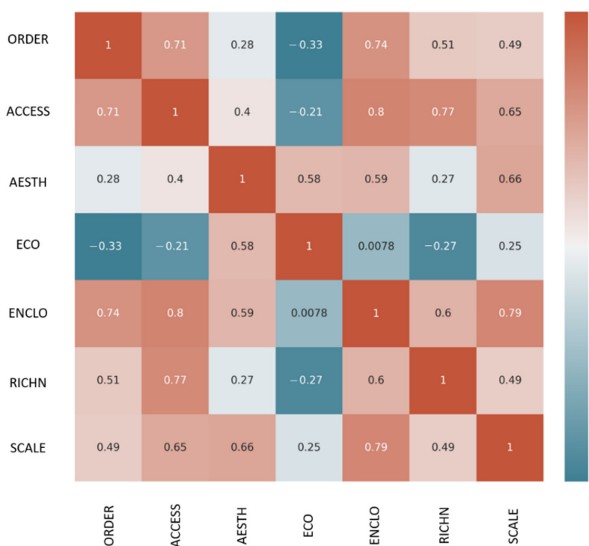

**Figure 7.** Pearson's correlation of 7 perceptions.

**Table 1.** Descriptive statistics of all variables.

| Variable | Syncopate | Description | Count | Mean | Std.Dev. | Min | Max | Data Source and Access Time |
|---|---|---|---|---|---|---|---|---|
| **Spatiotemporal Vitality attributes** | | | | | | | | |
| Vitality overal | AllHeat | The overall active population in all time periods | | 720.978 | 556.843 | 0 | 3319.38 | |
| Vitality morning | Mor_Heat | The morning active population | 101,035 | 220.042 | 158.938 | 0 | 1194.273 | Scraping from Baidu API and calculated in QGIS (2022) |
| Vitality noon | Noo_Heat | The noon active population | | 275.355 | 199.553 | 0 | 1351.23 | |
| Vitality evening | Eve_Heat | The evening active population | | 225.582 | 180.555 | 0 | 917.633 | |
| **Functional-based attributes** | | | | | | | | |
| Shannon_Wiener_Diversity | FDI | POI Functional Diversity | | 1.12 | 1.224 | 0.017 | 2.242 | |
| Functional Density | FDE | POI Functional Density | 39,375 | 26.82 | 12 | 1 | 414 | Scraping from Amap API and calculated in QGIS (2021) |
| Amenities reachability | ACR | Reachability from each segment to POIs | | 0.031 | 0.027 | 0.004 | 1.923 | |
| **Static Streetscape attributes** | | | | | | | | |
| Aesth_Score | AESHT | Perceived Aesthetic | | 0.394 | 0.39 | 0.228 | 0.697 | |
| Enclo_Score | ENCLO | Perceived Enclosure | 102,287 | 0.341 | 0.353 | 0.041 | 0.805 | Predicted by ML models with view indices extracted from SVIs (2022) |
| Richness_Score | RICHN | Perceived Richness | | 0.469 | 0.486 | 0.142 | 0.778 | |
| Scale_Score | SCALE | Perceived Human scale | | 0.383 | 0.376 | 0.175 | 0.794 | |
| **Through-movement probability attributes** | | | | | | | | |
| Choice/Betweenness 1 km | BET1k | Logarithm of Betweenness/Choice 1 km | 101,035 | 3.211 | 3.415 | 0 | 5.205 | Guangzhou Road Network Shapefile (2019) and calculated in Depthmap |
| Choice/Betweenness 5 km | BET5k | Logarithm of Betweenness/Choice 5 km | | 5.063 | 5.308 | 0 | 7.332 | |
| **Attributes interaction in dynamic models** | | | | | | | | |
| Aesth_Score * choice1k | AESTH_BET1k | Perceived Aesthetic through walking | | | | | | |
| Enclo_Score * choice1k | ENCLO_BET1k | Perceived Enclosure through walking | | | | | | |
| Richness_Score * choice1k | RICHN_BET1k | Perceived Richness through walking | | | | | | Predicted by ML models with view indices extracted from SVIs and multiply by Choice1km and Choice5km respectively, and normalized in the same sampled data points |
| Scale_Score * choice1k | SCALE_BET1k | Perceived Human scale through walking | 39,375 | | | | | |
| Aesth_Score * choice5k | AESHT_BET5k | Perceived Aesthetic through driving | | | | | | |
| Enclo_Score * choice5k | ENCLO_BET5k | Perceived Enclosure through driving | | | | | | |
| Richness_Score * choice5k | RICHN_BET5k | Perceived Richness through driving | | | | | | |
| Scale_Score * choice5k | SCALE_BET5k | Perceived Human scale through driving | | | | | | |

### 3.2.2. Through-Movement Probability Route Choice of Walking and Driving

Conventional studies used people's actual movement data from sensors or travel surveys to determine the variable according to different travel modes [16,17,40]. In this specific research, we argue that people's movement will be affected by the street view appearance, that is, the trajectories and GPS traces data was affected by the perception of the street scene, which means such trace data may have a certain collinearity when interacting with the street perception variables.

Arguably, in the network-based measurement method, space syntax, angular betweenness serves as an important indicator of movement potential measurement [41,62]. It has been proved that there is empirical scientific support (e.g., the capability of the street layout itself to predict pedestrian movement is presented in natural movement in the theory of spatial syntax. The angular segment analysis has a high correlation with the observed vehicular flow [42,45,70] of the high correlation between the choice/betweenness measurement and movement flow in different travel distances, representing the movement patterns of vehicles and pedestrians [71]. In this study, space syntax angular betweenness is used to compare the different potential factors between two types of transportation, walking and driving.

In the meantime, the road segment data is more accessible and less time consuming when compared to mining the trajectories data, which means this method has the potential to be extended to cross-scale urban regions.

It was also believed that urban forms and street patterns are generated by social forces [72] and the so-called "place" has different hierarchies in the network structure. In other words, the non-spatial stationary of opportunities of perceiving the surrounding environment through movement is the embodiment of regional economic emphasis [73]. For commercial places arranged on street sections with high movement potential of a certain traffic mode, its suitable street interface is more likely to be perceived by people in the corresponding traffic mode, thus enhancing people's positive perception of the region during the travel process.

Follow these lines, we chose 1 km as the walking probability distribution radius, from which it is about a 15-min walk [74], 5 km as the driving probability distribution radius, based on the average driving distance in a Guangzhou taxi report (4.7 km average distance), Guangzhou Robo-Taxi trial report (4.9 km average distance) and Guangzhou online-ride-hailing platform report (6.1 km average distance).

The measurement can be formulated by the Equation (1), as follows:

$$Betweenness(S_i) = \sum_{j=1}^{n} \sum_{k=1}^{n} \frac{P_{jik}}{P_{jk}} \tag{1}$$

where $P_{jk}$ indicates the shortest paths from $j$ to $k$, and $P_{jik}$ denotes the shortest paths from $j$ to $k$ passing through street segment $S_i$ [41,71]. The calculation of the walking and driving probability were computed at 1000 m and 5000 m metric radii in the DepthmapX software, Version 0.50 [75] and visualized with QGIS (ver. 2.18) software (Figure 8) (Table 1).

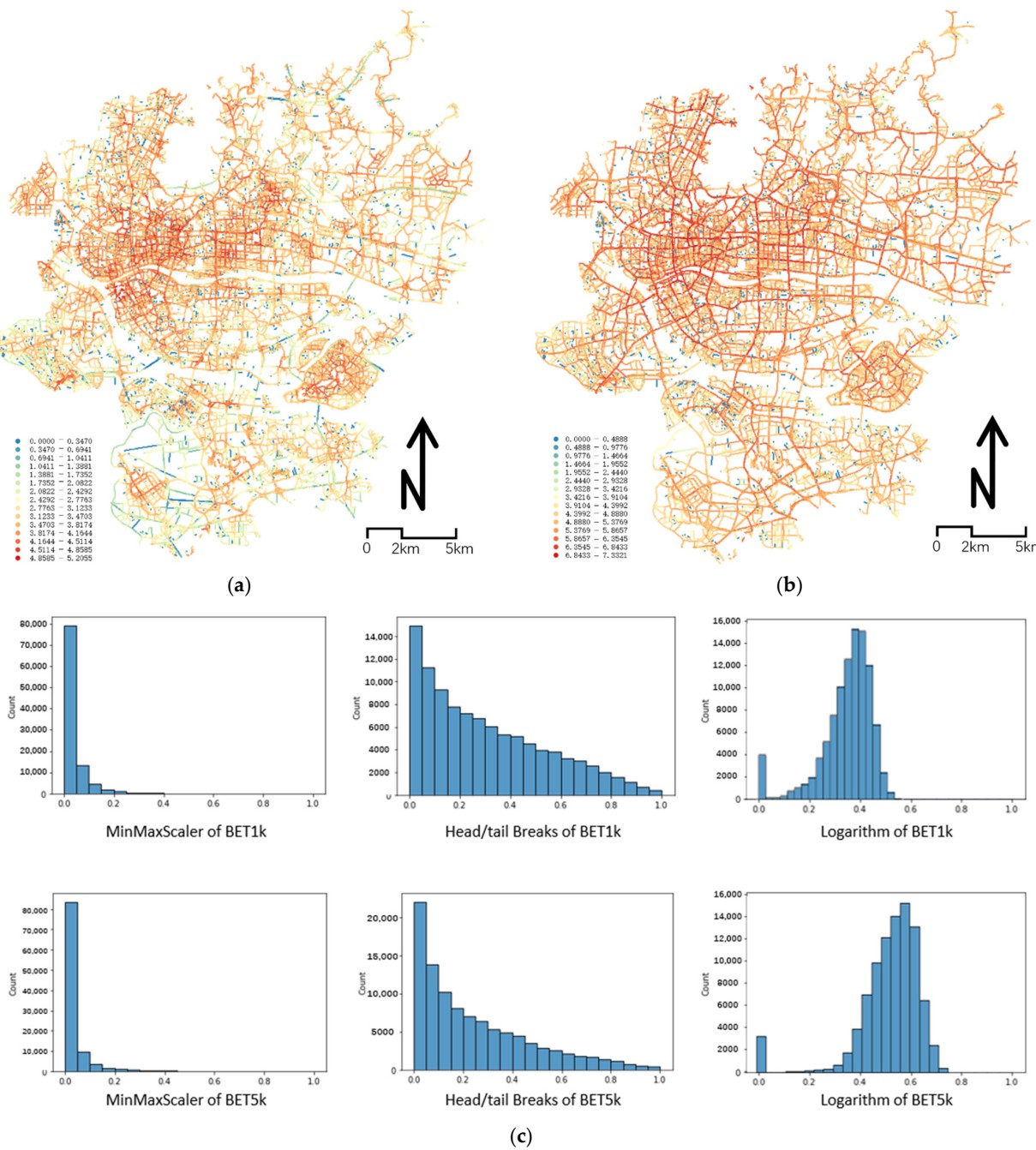

**Figure 8.** Spatial distribution of betweenness (movement potentials) at radii (**a**) Logarithm of Betweenness/Choice 1 km and (**b**) Logarithm of Betweenness/Choice 5 km and (**c**). Histogram of Betweenness/Choice at 1 km and 5 km in Central Guangzhou (N = 101,035). Normalized by MinMaxScaler, Head/tail Breaks and logarithm.

In terms of the distribution of data following a heavy head-tailed distribution, several normalization methods such as log normal distributions and head-tail breaks-classifier or Jenks clustering for standardization can be particularly effective [76,77].

### 3.2.3. Control Variables

From the previous study [48,69,78], by definition, urban functional condition requirements indicate that three principal aspects of urban function attraction variables—density, diversity and distance to the reachable complimentary amenities—are essential. Therefore, the function-based data (i.e., point of interests) were used to calculate the urban functional attraction variables, including the functional density, functional diversity and amenity clustering reachability (Table 1). The POIs data in the year of 2021 (717,740 records in total) was downloaded from Amap and spatial analysis in GIS. 125,703 records of POIs data in the central area of Guangzhou were collected and separated into 12 living commercial and amenities clusters: cafes, restaurants, street retail, shopping malls, culture facilities, sport facilities, recreation facilities, tourist attractions, hotels, hi-tech centers, offices and medical institutions.

Functional density (Figure 9a) was measured through the total quantities of the POIs within the given radius buffer of each hexagon grid centroid [48]. In this study, we selected 300 m radii for density calculation based on the 5 min' walk life circle proposal in China. Functional diversity (Figure 9b) can be measured in several popular ways, such as the dissimilarity index method [38] and the Shannon entropy method [69,78,79]. In this study, Shannon diversity index is applied to measure the diversity of 12 categories of urban points of interest from street segments within the hexagon grid *i* with radius *r* = 300. The computation can be calculated formally as follows (2):

$$\textit{Functional Diversity}_{(i,r)} = -\sum\nolimits_{k=1}^{k} p_{(i,r)} \times \ln p_{(i,r)}, \; \{\text{dist}(ij) < r\} \tag{2}$$

The amenities reachability (Figure 9c,d) measures the closest distance to all the reachable urban activities from each street segment within a given radius. This variable reveals the accessible efficiency of the urban amenities from all the places within the street. Based on the delivery efficiency model [69,78], the method distance to nearest hub in QGIS was used to calculate the distance from each point j with 50 m spacing, divided on the street segments to all its closest facilities within the metric radius *r* = 300. $N_{(i,r)}$ is the number of reachable links from road segments to POIs at the same radius. The result was accumulated in each grid *i*. The index can be represented as the following Equation (3):

$$\textit{Amenities Reachability}_{(i,r)} = N_{(i,r)} / \sum\nolimits_{j=1}^{j} \textit{Hub Distance}_j, \; \{dist(ij) < r\} \tag{3}$$

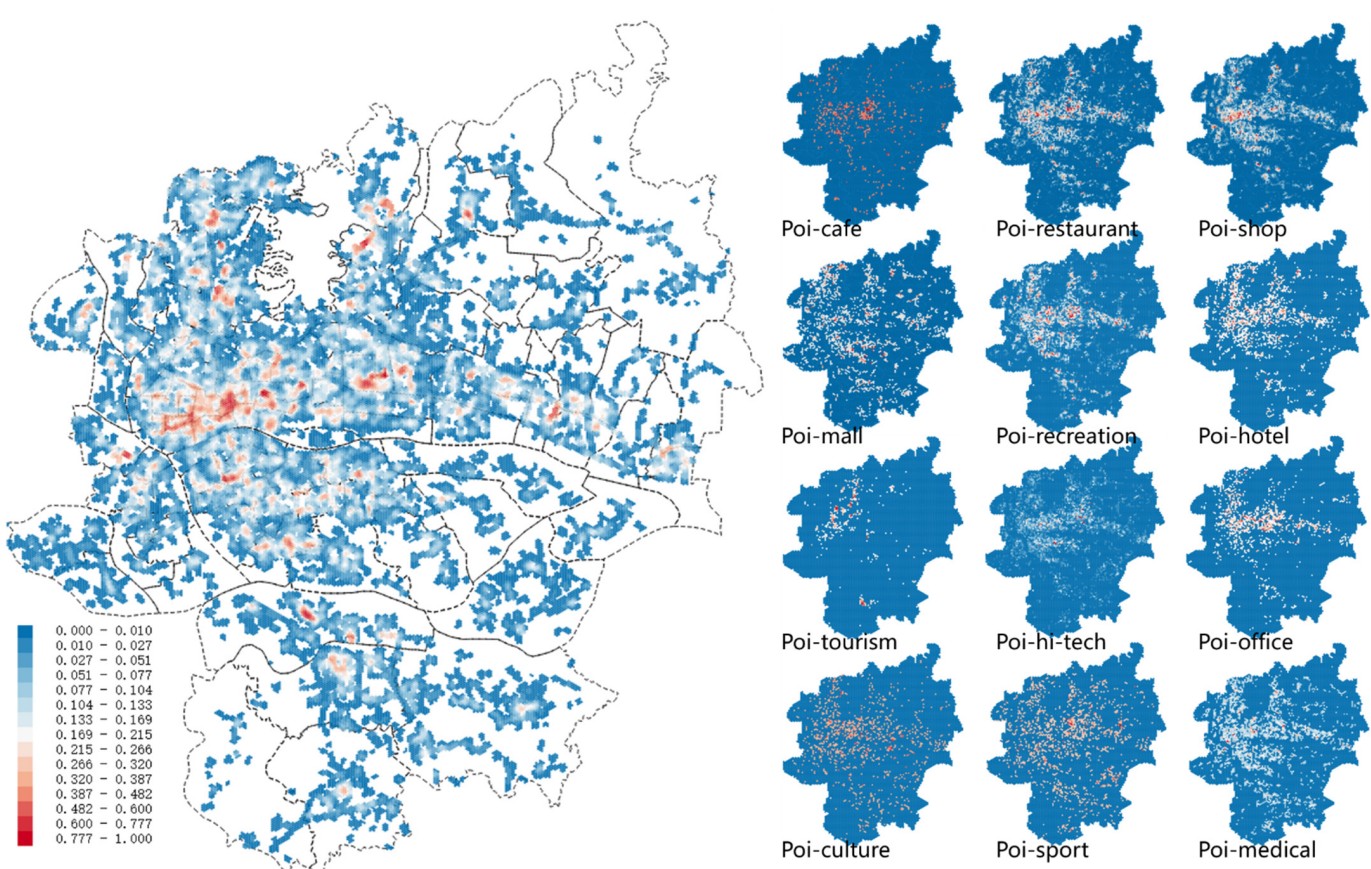

(**a**) Functional Density for 12 categories altogether and separately

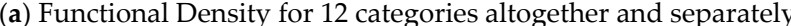

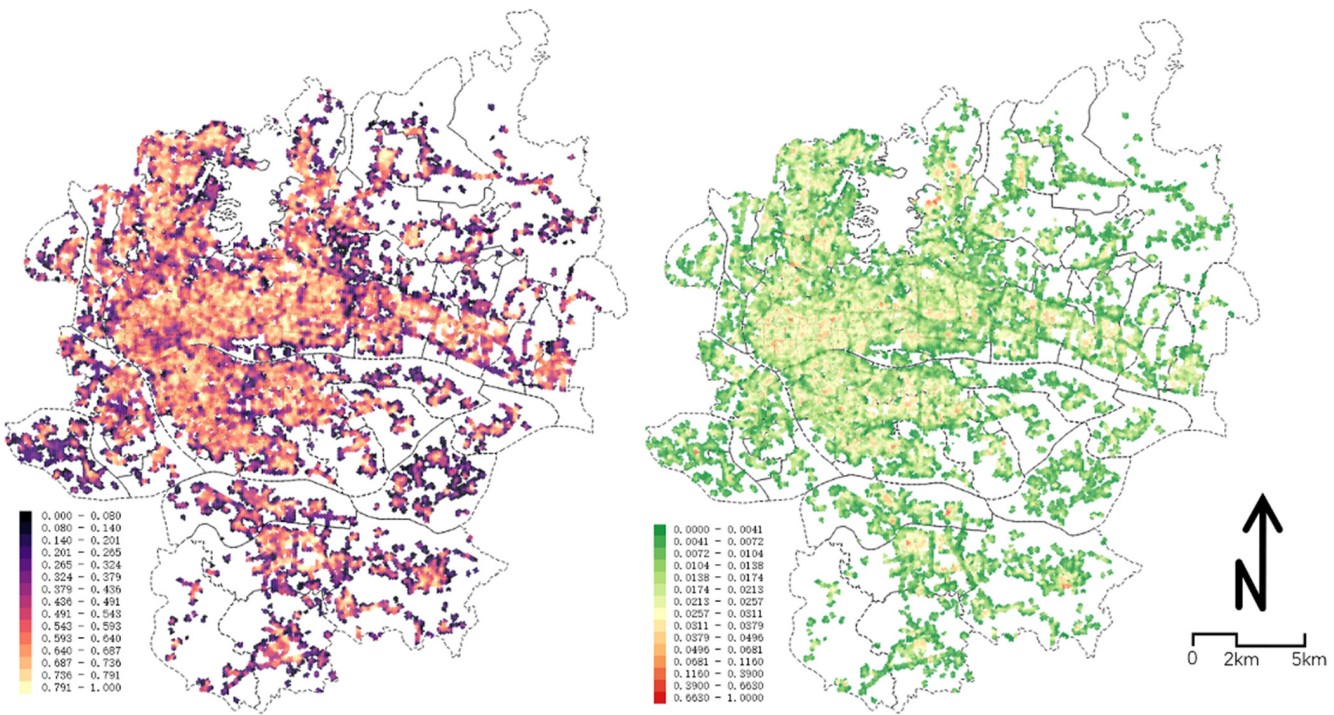

(**b**) Functional Diversity (Shannon Diversity Index)    (**c**) Overall Amenities Reachability

**Figure 9.** *Cont.*

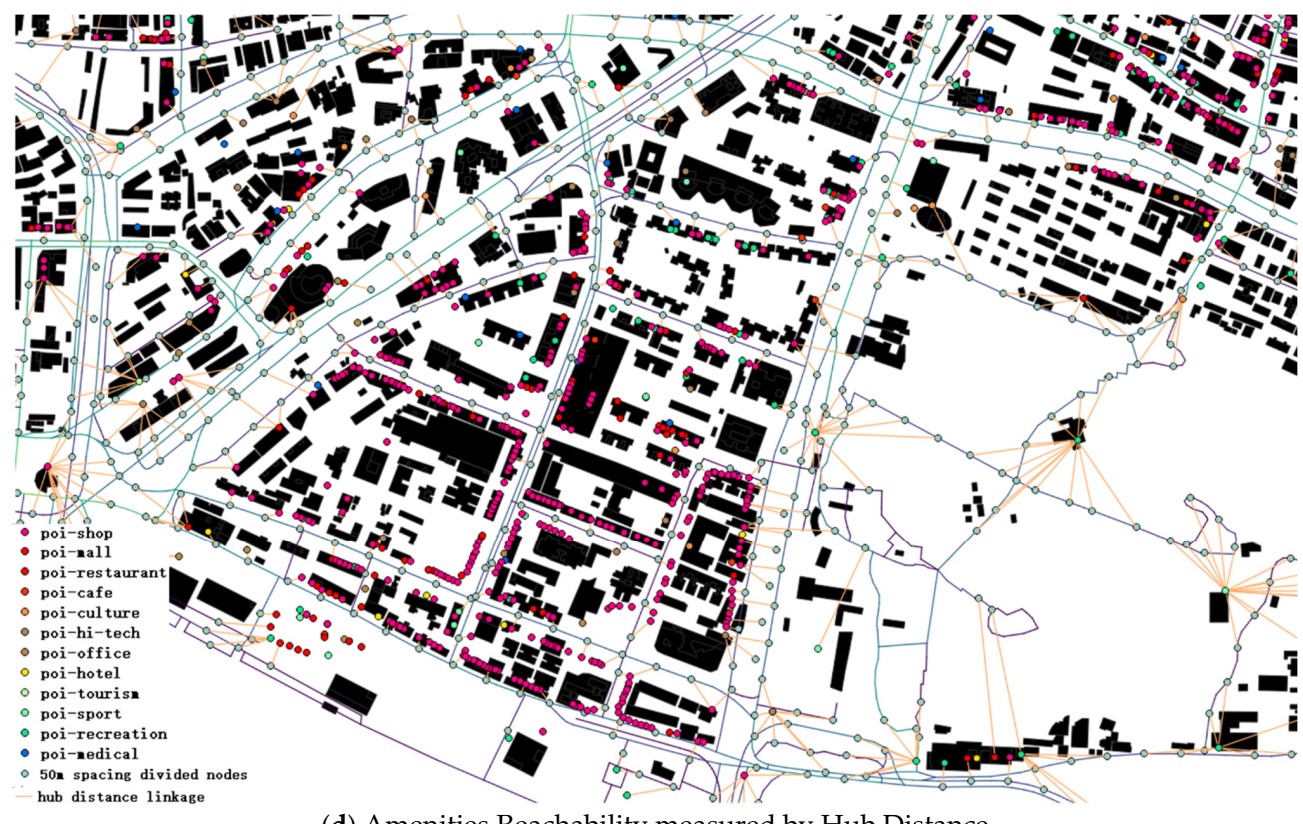

(**d**) Amenities Reachability measured by Hub Distance

**Figure 9.** (**a**) Functional density, (**b**) Functional diversity, (**c**) Amenities reachability, and (**d**) Amenities reachability measured by hub distance.

### 3.2.4. Dependent Variable

There are several ways for measuring street vitality. The most representative methods, for example, can be to set up a live-stream camera to capture the pedestrians' activities [10], using the walk-by observation method to record social activities and pedestrian counts [11] and using urban perception scoring compared on the basis of different street images for the relative vitality relationship in the city [3]. These methods are either inapplicable at large urban scales or unable to represent the vitality without street view data.

Considering the importance of time diversity (i.e., attracting people at different times of the day) and space diversity (i.e., attracting people from different urban districts) [3,37,60] being presented in the urban scale, in this study, street vitality (Table 1) was measured by using the active hotspot of the flow on the street at different times in a day. The heatmap in Guangzhou was collected from Baidu (http://api.map.baidu.com/lbsapi/creatmap/, accessed on 16 June 2022)-the largest navigation service and travel behavior data platform in China with a specific API and Python script for mapping the distribution of the mobile phone check-in location (https://developer.baidu.com/map/, accessed on 16 June 2022).

We first recorded the active hotspot every 2 h from 8:00 to 22:00 on a workday.

Second, a parameter transfer method was used to calculate the active population from the heatmap. The measurement can be formally represented as the following Equation (4) [80]:

$$P_i = \begin{cases} \frac{10-0}{151-60} \times (SA_i - 60), & 60 \leq SA_i \leq 151 \\ 10 + \frac{20-10}{163-151} \times (SA_i - 151), & 151 < SA_i \leq 163 \\ 20 + \frac{40-20}{170-163} \times (SA_i - 163), & 163 < SA_i \leq 170 \\ 40 + \frac{60-40}{179-170} \times (SA_i - 170), & 170 < SA_i \leq 179 \\ 60, & 179 < SA_i \leq 194 \end{cases} \; ; \; Z_j = \sum_{j,i}(S_z P_{j,i}) \qquad (4)$$

$P_i$ is the population density per ha of the grid $i$ (50 m by 50 m). $SA_i$ is the Alpha channel value of the grid measured by the Baidu Heatmap based on the mobile phone check-in data. The number of population activities on each pixel can be obtained by the pixel size $S_z$ and the grid population aggregation density $P_i$, and then the number of population activities in the grid $Z_j$ can be obtained by summing the population number of each pixel in the grid. By using the equation above, we can transfer the Baidu Heatmap color channel into actual population density in different time periods, which proved to be reliable in previous research [80–83]. Third, these 8-h vitality data were accumulated and divided into three groups, namely morning (8 a.m., 10 a.m., 12 a.m.), noon (0 p.m., 2 p.m., 4 p.m.) and evening (6 p.m., 8 p.m., 10 p.m.) vitality (Figure 10a). This is shown in the equation below, where VT (i.e., vitality time diversity) represents the dynamic vitality change in different hours in grid $i$ (5) and VS (i.e., vitality space diversity) represents the different distribution of instant vitality at each time in grid $i$ (6). $n$ is the time in three periods, in which $n = 10$ a.m. in the morning, $n = 2$ p.m. in the noon and $n = 8$ p.m. in the evening.

$$VT_{(i,n)} = \left| population_{(n,i)} - population_{(n-2,i)} \right| + \left| population_{(n+2,i)} - population_{(n,i)} \right| \quad (5)$$

$$VS_{(i,n)} = population_{(n-2,i)} + population_{(n,i)} + population_{(n+2,i)} \quad (6)$$

We argued that both vitality time diversity and space diversity should be considered. Vitality morning, vitality noon and vitality evening, were calculated using the equation in (7):

$$Vitality_{(i,n)} = VT_{(i,n)} + VS_{(i,n)};\ \{morning\ (n = 10\ am);\ noon\ (n = 2\ pm);\ evening\ (n = 8\ pm)\} \quad (7)$$

We accumulated the overall vitality by summing the $VT$ and $VS$ in the three time steps (8) (Figure 10b).

$$Vitality\ overall = Vitality\ morning + Vitality\ noon + Vitality\ evening = VT_{(i,10am)} + VS_{(i,10am)} + VT_{(i,2pm)} \\ + VS_{(i,2pm)} + VT_{(i,8pm)} + VS_{(i,8pm)} \quad (8)$$

We took Vitality overall as the dependent variable (Y) in baseline model, Model1, 2 and 3; Vitality morning, Vitality noon and Vitality evening as the dependent variables in Model3M, 3N and 3E, respectively (Tables 1 and A1).

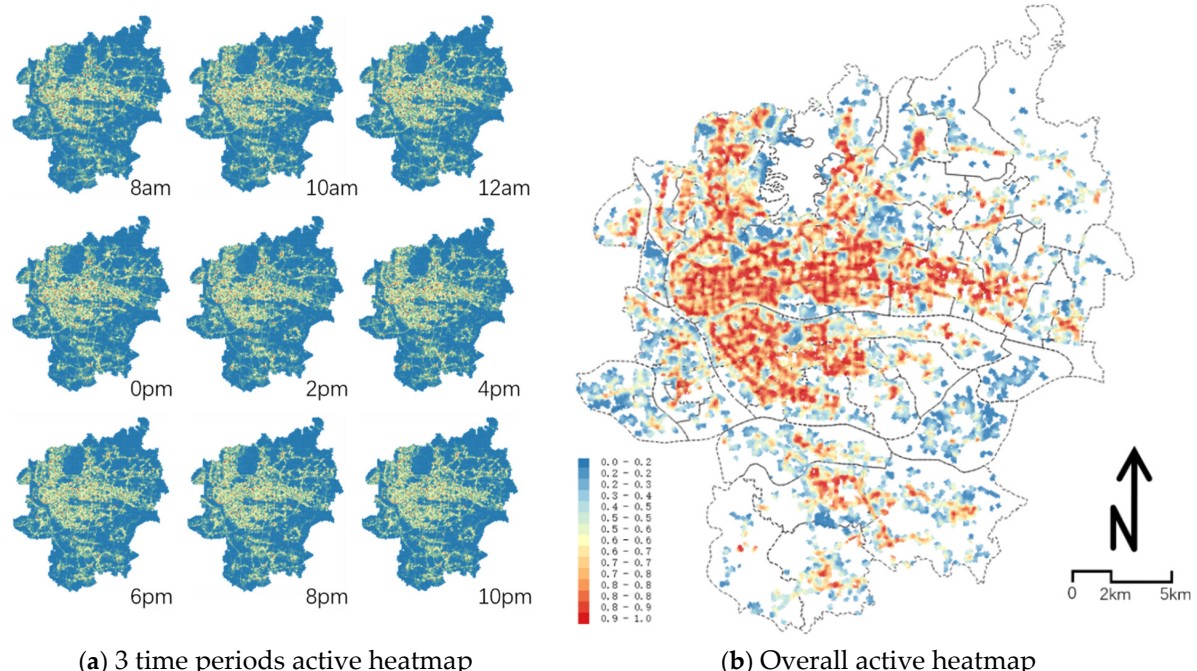

(**a**) 3 time periods active heatmap　　　　　　　　　　　(**b**) Overall active heatmap

**Figure 10.** Demonstration of the spatial and time diversity distribution of street vitality.

### 3.3. Model Architecture

To investigate how different movement modes affects perceptions and how such interaction affects vitality, we considered the comparison of classic static perception-vitality model and the interaction model, in which the two major variables streetscape perception and through-movement probability for walking and driving interact. There are, however, several different approaches for incorporating the travel probability into the model (e.g., the relative high-low hierarchy of accessibility) [48,49]. For this specific task, we used the movement probability index as a weight on all types of perception within the streets [84]. Therefore, the through-movement perceptions were introduced as the new variables in the dynamic models. The dynamic perception can be measured through the assumption (9):

$$Dynamic\ perception(P_{i,\ r,k}) : \sum_{j=1}^{n} Betweenness_{jr} \times \sum_{j=1}^{n} Perception_{jk} \tag{9}$$

where $j$ was the equally divided nodes on the network segments in 200 m sampling grid $i$. Perception $k$ was the 7 types of perception. Radius $r$ was the assumed movement distance of 1 km and 5 km, which represent two different travel modes.

We employed the OLS regression to set up four stepwise models. First, we added each set of attributes into the separated OLS models to be aware of their explanatory power. The standard baseline model—the initial one without any perception attributes—was constructed by only using control variables (10):

$$Y_v = β_0 + β_1 FDI + β_2 FDE + β_3 ACR + ε \tag{10}$$

where $Y_v$ is the predicted value of the dependent variable, *FDI*, *FDE*, *ACR* are the control variables: *Functional Density i*, *Functional Diversity(i,r)* and *Amenities Reachability I*, respectively, and $β_0$, $β_1$, $β_2$ and $β_3$ are the coefficients to be estimated through our models.

Second, based on the baseline model, we added static perception-Model 1 (11):

$$Y_v = β_0 + β_1 FDI + β_2 FDE + β_3 ACR + β_4 Static\ Percep + ε \tag{11}$$

and dynamic perception (i.e., through-movement walking and driving perceivable perception)-Model 2 (12):

$$Y_v = β_0 + β_1 FDI + β_2 FDE + β_3 ACR + β_4 Dynamic\ Percep + ε \tag{12}$$

where *Static Percep* was the *Static Perceptions(Pi,r,k)* including AESTH, ENCLO, RICHN and SCALE, and *Dynamic Percep* was *Dynamic Perceptions(Pi,r,k)* including AESTH, ENCLO, RICHN and SCALE with both BET1k and BET5K weighted, respectively, to compare their different strengths.

Third, the fourth model was the interaction model, considering both static and dynamic perception contributing to urban vitality-Model 3 (13) (Table 2):

$$Y_v = β_0 + β_1 FDI + β_2 FDE + β_3 ACR + β_4 Static\ Percep + β_5 Dynamic\ Percep + ε \tag{13}$$

**Table 2.** Model comparison.

| | Selected Model | | Adjusted R Square | Std Error of the Estimate | AIC | N |
|---|---|---|---|---|---|---|
| Baseline | All day without perception data | | 0.381 *** | 0.003 | −16,610 | 31,526 |
| Model 1 | All day with static perception | | 0.439 *** | 0.005 | −19,680 | 31,526 |
| Model 2 | All day with dynamic perception | walking | 0.442 *** | 0.003 | −19,860 | 31,526 |
| | | **driving** | **0.450 ***** | 0.003 | −20,300 | 31,526 |
| Model 3 | All day with static and dynamic interaction model | walking | 0.453 *** | 0.005 | −20,500 | 31,526 |
| | | **driving** | **0.470 ***** | 0.005 | −21,480 | 31,526 |

Notes: *** stand for significance level (*p* value) < 0.01. The travel modes with higher R Square value are bolded.

Lastly, since many variables tend to be closely correlated, multicollinearity might exist among variables of human movement patterns and multiple indices of human perception [4]. The variance inflation factor (VIF) test was used to check for variables with correlation problems (Pearson pair plot value > 0.8 and VIF > 10) [6]. Before running the OLS model, we removed the variables which were unimportant and achieved significant multicollinearity with other important factors [27,85]. Hence, 3 variables were removed from our model, namely order score, access score and eco-score. In terms of the new dynamic perception variables in the dynamic model that might have high multicollinearity problems, the *p*-value for interaction variables was not affected by the multicollinearity [86] (Table 3).

**Table 3.** Comparison of the dynamic perception model of walking and driving with the variables of VIF < 10.

| Variable | VIF | Model3-Walking | | Model3-Driving | |
|---|---|---|---|---|---|
| **Intercept** | **27.62** | **Coefficient** | **Std Err** | **Coefficient** | **Std Err** |
| FDI | 1.64 | 0.35 *** | 0.006 | 0.33 *** | 0.006 |
| FDE | 1.66 | 0.64 *** | 0.014 | 0.60 *** | 0.014 |
| ACR | 1.60 | −0.7 *** | 0.092 | −0.60 *** | 0.091 |
| AESTH | 2.52 | −0.31 *** | 0.02 | −0.38 *** | 0.018 |
| ENCLO | 4.44 | 0.13 *** | 0.024 | 0.30 *** | 0.021 |
| RICHN | 2.16 | 0.28 *** | 0.017 | 0.27 *** | 0.015 |
| SCALE | 4.89 | −0.03 | 0.032 | −0.05 * | 0.027 |
| BET1k | 1.90 | | | | |
| BET5k | 1.96 | | | | |
| AESTH_BET1k | | −0.27 *** | 0.057 | | |
| ENCLO_BET1k | | 0.03 | 0.066 | | |
| RICHN_BET1k | | 0.53 *** | 0.037 | | |
| SCALE_BET1k | | −0.16 ** | 0.084 | | |
| AESTH_BET5k | | | | −0.07 * | 0.056 |
| ENCLO_BET5k | | | | −0.26 *** | 0.065 |
| RICHN_BET5k | | | | 0.67 *** | 0.036 |
| SCALE_BET5k | | | | −0.06 | 0.078 |
| Adjusted R square | | 0.453 | | 0.47 | |
| AIC | | −20,500 | | −21,480 | |
| BIC | | −20,400 | | −21,380 | |

Notes: ***, **, * indicate *p* > |t| at significant levels 0.001, 0.05 and 0.1 respectively.

## 4. Results and Discussion

### 4.1. OLS Results

4.1.1. Verifying Dynamic Perception and Static Perception

As shown in Table 2, adding the static perception variables can significantly improve the baseline model from 0.381 to 0.439 of the R square (Table 2).

More importantly, Model 2 and Model 3, where dynamic perception factors were involved, have better explanatory power than the model only with static perception factors. Table 2 shows the coefficients of either the dynamic perception through walking (1 km movement potential weighted) or driving (5 km movement potential weighted) have larger impacts on vitality than static perception, with the improvement in the R square of 0.442 and 0.450 for walking and driving, respectively, in Model 2. Since Model 3 considers the interaction factors of both static perception and dynamic perception, the result shows even better in the R square with 0.453 and 0.470 for walking and driving, which indicate that both static perception and dynamic perception contribute to the street vitality simultaneously.

### 4.1.2. Evaluating Walking and Driving Through-Movement Perception

First, in terms of the performance of the two modes of through-movement perception, the walking's $R^2$ is 0.442 in the dynamic perception model and 0.453 in the interaction model, while the driving perception performs better in these two models with 0.008 and 0.017 greater of $R^2$ than walking (Table 2).

Second, according to the results of variables coefficient in (Table 3), factors 5 and 6, which represent the "enclosure" and "richness" spatial perception under the walking and driving modes, both have positive effects on street vitality. However, an interesting phenomenon occurred in the interaction model (Model 3), in which the sense of enclosure of driving had a negative impact on vitality while the walking sense of enclosure still had a positive effect.

Third, most of the dynamic perception variables were significantly associated with vitality, with 0.000 in *p*-Value in Model2, and in Model3, except for the sense of enclosure when walking, and the sense of aesthetic and human scale when driving (Table A1).

Lastly, the overall fitness of the R square of driving perception was greater than the walking counterpart in each dynamic model (Model 2, Model 3), which means the driving mode of streetscape perception can better illustrate the overall and dynamic vitality pattern of Guangzhou. In other words, from a development mode and urban governance perspective, the existing development principle is more likely to be auto-oriented traffic policies for Guangzhou in the city scale.

### 4.1.3. Overall Comparison with Three Time Periods and Detailed Variables

In terms of the time dimension, the intention of our proposed model was to test the change of the coefficient strength in three different time periods (Table A1). The outcomes from the morning and evening periods outperformed the noon period in both walking and driving modes. At the same time, the three control variables performed in a stable manner in the coefficient in three time periods, from which can be understood that they follow the relationship we built between perception and vitality in the noon period to a lesser extent. People tend to move and stay in a place where they can have a more impressionable sense of place in the morning and evening rather than in noon. For example, in the morning, the workplaces where people commute to are more likely to be in the areas with a better sense of place. At the same time, groceries, other shops and restaurants become more popular in the night economy and most of them are located in the areas with strong streetscape perception.

The perception variables, especially richness and human scale, performed variously in three models. First, the detail can be found that in each dynamic model of the different time periods; the dynamic richness perception always made a positive contribution to the fitness of vitality. In contrast, the dynamic human scale perception variable always played a negative role in each model. Second, all six dynamic richness attributes were ranked among the three most influential factors, which is in accordance with the active diversity and time diversity in the literature [37,60]. Third, on the flip side, the impact of the human scale variable was not positively correlated with vitality, as shown in the previous studies [27,36], which was supposed to reveal that the small-scale streetscape appearance might often be neglected by the urban developers in the development of Guangzhou.

### 4.2. Comparison of Related Studies

When compared with related studies [6,68], we build on existing analytical methods that have been validated, while considering human perception of the street under different modes of transportation. On the other hand, our study is more scientifically quantifiable compared to the non-numerical variables (high and low) employed in the analysis of accessible green by [48,49]. Thus, we argue through a case study that the dynamic perception of driving in central Guangzhou is more relevant to the overall street vitality distribution. Our study complements the lack of consideration of bias in through-movement in related field studies.

## 5. Conclusions

In this paper, we have presented a new approach to the application of through-movement perception as a dynamic perspective to explain the street vitality and different influences on perception through walking and driving. Thus, we argue that in central Guangzhou, the dynamic perception of driving was more effective in explaining street vitality. We also expect to capture the shared experiences in which better dynamic perceptions can engender more lively streets and regions, and thus our approach may represent a stepping stone in bridging the relationship between a static sense of place and the dynamic movement potential of two travel modes.

### 5.1. Complementary Effects between Two Modes

Static streetscape perception indexes can still be used as a supplement to dynamic perceptions. First, it is difficult to thoroughly simulate the dynamic travel process of humans, since perceptions are multifaced [6]. While static street perception gives all places an equal opportunity to be perceived, it is relatively stable. Secondly, if the dynamic perception measurement is applied to a single nuclear city, where high street perceptions are similarly distributed to the high movement potential, it may increase the collinearity of variables in the dynamic model, limiting the measurement credibility of this method. Therefore, the use of both static and dynamic perception methods can increase the overall model credibility and explanatory power to a certain extent. At the same time, the dynamic perception of the two travel modes can be used not only as independent objects to compare the development orientation of the regions, but also as a complementary verification means to judge the credibility of prediction.

### 5.2. Implications for Urban Planning

Our study offers both insights and tangible applications in urban planning, informing urban revitalization for commercial developers, policy makers, researchers and planners. First, our study shows that the distribution of better dynamic perception is intermittent in Guangzhou, and there are even some continuous areas of high vitality with a low relationship of streetscape perception. In other words, there are some specific places (e.g., the street space along the viaduct) whose streetscape quality is not proportional to the high activity hotspot, which has become a blind spot in shaping better urban scape.

Second, our study helps policy makers to better plan and improve urban transportation facilities to meet the needs of various commercial facilities and citizens, which might lead to more responsive urban planning and regional commercial development. For example, commercial investors get more customers from the high street vitality generated by the surroundings for their suitable travel modes, while citizens enjoy better street quality and increase the potential to work and to purchase property nearby, which brings more tax revenue for the government. Third, the data-driven methods in this study can help researchers understand human perception and travel choices, to better approximate the interaction between humans and the environment from the perspectives of visual and spatial movement potential.

### 5.3. Limitations

Firstly, although the actual observation results of traffic flow in different modes are close to the simulation of walking and driving movement modes by using different distance radii of choice index in space syntax, the distribution of population and fixed travel destinations is not spatially uniform. Therefore, this method cannot completely cover the real dynamic flow patterns.

Secondly, taking different transport means is closely related to the flow of segregation in different income, occupation and preference, and these features also play a conclusive impact on where they go and behave, which was the missing part in this study.

Thirdly, there may be a mutually exclusive relationship between different modes of transportation, for example, the high traffic flow of vehicles will block the movement of pedestrians and shape the walking patterns to avoid such high-level major collectors.

Lastly, we should of course be aware that space syntax or any other network theory based on computational models can be helpful in studying the likelihood of social behaviors but cannot precisely predict the movement flow of people. Humans are creative and their actions cannot be explained solely based on cause-effect (i.e., stimulus-response) mechanisms. In other words, people cannot be reduced to 'causal systems' [87,88]. This research is based on the view that space is a part of the development of social and economic activities [63], that is, the source of influence of urban and street structure on social activities in urban streets is the shaping of urban street structure and appearance through the interaction of people's social activities. However, if in the modern anthropocene, the material space no longer guides human activities, information flow and capital flow and the logistics in the operation of the city are no more dependent on the city and street structure itself, the correlation between the material space and human social activities will be reduced.

**Author Contributions:** Conceptualization, Y.W. and W.Q.; methodology, Y.W. and W.Q.; software, Y.W., W.Q. and W.L.; validation, Y.W., W.Q., Q.J. and W.L.; formal analysis, L.D.; investigation, Y.W.; resources, Q.J., L.D. and T.J.; data curation, Y.W. and W.L.; writing—original draft preparation, Y.W.; writing—review and editing, W.Q., Q.J., L.D. and T.J.; visualization, L.D.; supervision, Y.W. and W.Q. All authors have read and agreed to the published version of the manuscript.

**Funding:** This research received no external funding.

**Data Availability Statement:** The study did not report any publicly archived datasets.

**Conflicts of Interest:** All the authors did not receive any research grants and declare no conflicts of interest including financial interests (membership, employment, consultancies, stocks/shares ownership, honoraria, grants or other funding) and non-financial interests (personal or professional relationships, affiliations, personal beliefs). No sponsors had any role in the design, execution, interpretation, or writing of the study.

## Appendix A

**Table A1.** Model comparison.

| Selected Model | | | Adjusted R Square | Std Error of the Estimate | $p > |t|$ (Sig.) | AIC | N |
|---|---|---|---|---|---|---|---|
| OLS Regression Analyses | | | | | | | |
| Baseline | All day without perception data | | 0.381 | 0.003 | 0.000 *** | −16,610 | 31,526 |
| Model 1 | All day with static perception | | 0.439 | 0.005 | 0.000 *** | −19,680 | 31,526 |
| Model 2 | All day with dynamic perception | walking | 0.442 | 0.003 | 0.000 *** | −19,860 | 31,526 |
| | | **driving** | **0.450** | 0.003 | 0.000 *** | −20,300 | 31,526 |
| Model 3 | All day with static and dynamic interaction model | walking | 0.453 | 0.005 | 0.000 *** | −20,500 | 31,526 |
| | | **driving** | **0.470** | 0.005 | 0.000 *** | −21,480 | 31,526 |
| Model 3M | Interaction model with morning vitality | walking | 0.447 | 0.005 | 0.000 *** | −19,950 | 31,526 |
| | | driving | 0.464 | 0.005 | 0.000 *** | −20,940 | 31,526 |
| Model 3N | Interaction model with noon vitality | walking | 0.425 | 0.005 | 0.000 *** | −19,280 | 31,526 |
| | | driving | 0.444 | 0.005 | 0.000 *** | −20,340 | 31,526 |
| Model 3E | Interaction model with evening vitality | walking | 0.440 | 0.005 | 0.000 *** | −22,910 | 31,526 |
| | | driving | 0.452 | 0.005 | 0.000 *** | −23,580 | 31,526 |

Notes: *** stand for significance level (*p* value) < 0.01. The travel modes with higher R Square value are bolded.

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
