# Peer review of "Drivers or Pedestrians, Whose Dynamic Perceptions Are More Effective to Explain Street Vitality? A Case Study in Guangzhou"

_remotesensing, doi:10.3390/rs15030568_

Round 1

Reviewer 1 Report

I have no suggestions.

Author Response

Thank you for reviewing

Reviewer 2 Report

The author has done a very systematic study, especially the data analysis. The quality of the paper is very good, but I have some suggestions which I hope can help the author improve the quality of the paper.

1. The structure of Introduction and literature review is very strange. What problem does the author want to express in the preface? It seems to me that the things you're trying to express are actually based on a review of the literature.

2. In Figure 1, references can be omitted.

3. In the data part, the author uses a large amount of data. It is suggested that the author use a table to summarize the data, including the source, resolution and access time of different data.

4. In Part 3.2, the study framework can be placed after the study method.

5. In Figure 9, the different categories of POI data should be marked, in addition, the legend should be marked in part d.

6. Where are the formula and usage of OLS and GWR?

7, The authors did not focus on analyzing the similarities and differences between this study and other studies in the discussion part.

8. The authors' organization in the discussion and conclusion sections is strange, and it is suggested that the authors make adjustments in these two sections.

9. There is so much content in every part of the paper that it is hard to tell where the author's priorities lie. For some unimportant content, the authors could not write so much, especially in the part of experimental design.

10. “Drivers or Pedestrians, whose dynamic perceptions are more effective to explain street vitality? A case study in Guangzhou”. From the analysis results of the author, the author did not answer this result directly, including in the abstract and conclusion.

Other minor issues:

1. Line 421 Error!

2. Some fonts in the paper are inconsistent

Author Response

Thanks for the careful review, we have carefully revised and hope we have addressed all your concerns. We've learnt a lot through this process. Many thanks again. Please see attached word document.

Reviewer 3 Report

This paper suggests a new method for evaluating human perception of street views in relation to a network-based movement potential model. It establishes the link between dynamic through-movement perception and street vitality, which can serve as a foundation for further investigation in this area. This is a well-written paper that deserves publication. I recommend this paper can be published with some minor revisions.

1. Figure 1 reveals the relationship between street vitality and other elements. To better convey information through figures, please add annotations on the meanings of solid and dashed arrows.

2. Figure 3: In terms of format, please shorten the text inside boxes. The font used for the references could be smaller and the font of non-quoted text could be more consistent. OSL should be OLS, please check. In terms of content, OLS is followed by GWR in the figure, but GWR is not mentioned in the article. For the application of methods such as OLS and GWR, please add the formulas in the article. In addition, although the figure plots the idea of separately considering local scale and city scale, the article does not specifically analyze for them.

3. Please clarify Figure 4 and Figure 8 (c) in the revision. Figure names could be aligned and text in the figure could be resized. Please revise Figure 6: Histograms could be resized to avoid the images covering the border line, and axis titles could be revised. Please add annotations of 12 categories in Figure 9 (a) and legends of circles and lines in Figure 9 (d).

4. What significance does Figure 7 have for perception scores? Please explain. In addition, high correlations between two independent variables may reduce the prediction accuracy. However, the value between richness and enclosure is 0.79.

5. Formula (2), (3): Please check the letter cases. 

6. Page 18, Line 447 to 459: It may not be appropriate to use "+" to denote compositions in models. It is advised that the form be altered to avoid ambiguity and highlight differences among models.

7. Table 2: Why is there a column N and why is N 31526? 

8. Some clerical and grammatical mistakes. R2 should be R2; Page 13, Line 374: Figure.9c&d; Page 16, Line 415: 12a.m. in noon could be 0p.m, which consists with Figure 10(a); The layout of Figure 10 should be revised; Page 20, Line 478: Table x. Please check.

Author Response

Thank you dear Reviewer 3, we've largely revised the manuscript, hope this version is more straightforward, accurate, clean and effective. Best regards

Round 2

Reviewer 2 Report

Accept in present form